# ExoS effector in *Pseudomonas aeruginosa* Hyperactive Type III secretion system mutant promotes enhanced Plasma Membrane Rupture in Neutrophils

**Arianna D. Reuven, Sarah Katzenell, Bethany W. Mwaura, James B. Bliska**[ID]*

Department of Microbiology and Immunology, Geisel School of Medicine at Dartmouth College, Hanover, New Hampshire, United States of America

* james.bliska@dartmouth.edu

## Abstract

*Pseudomonas aeruginosa* is an opportunistic pathogen responsible for airway infections in immunocompromised individuals, including those with cystic fibrosis (CF). *P. aeruginosa* has a type III secretion system (T3SS) that translocates effectors into host cells. ExoS is a T3SS effector with ADP ribosyltransferase (ADPRT) activity. ExoS ADPRT activity promotes *P. aeruginosa* virulence by inhibiting phagocytosis and limiting oxidative burst in neutrophils. The *P. aeruginosa* T3SS also translocates flagellin, which can activate the NLRC4 inflammasome, resulting in: 1) gasdermin-D pores, release of IL-1β and pyroptosis; and 2) histone 3 citrullination (CitH3), nuclear DNA decondensation and expansion into the neutrophil cytosol with incomplete NET extrusion. However, studies with *P. aeruginosa* PAO1 indicate that ExoS ADPRT activity inhibits the NLRC4 inflammasome in neutrophils. Here, we identified an ExoS+ CF clinical isolate of *P. aeruginosa* with a hyperactive T3SS. Variants of the hyperactive T3SS mutant or PAO1 were used to infect neutrophils from C57BL/6 mice that were wildtype or engineered to have a CF genotype or defects in inflammasome assembly. Responses to NLRC4 inflammasome assembly or ExoS ADPRT activity were assayed and found to be similar for C57BL/6 or CF neutrophils. ExoS ADPRT activity in the hyperactive T3SS mutant regulated inflammasome, nuclear DNA decondensation and incomplete NET extrusion responses, like PAO1, but promoted enhanced CitH3 and plasma membrane rupture (PMR). Glycine supplementation inhibited PMR by the hyperactive T3SS mutant, suggesting ninjurin-1 is required for this process. These results identify enhanced neutrophil PMR as a pathogenic activity of ExoS ADPRT in hypervirulent *P. aeruginosa*.

**Data availability statement:** All data that underlie the findings of this study are publicly available from Zenodo (Zenodo. org) with the identifier: https://zenodo.org/records/14893677.

**Funding:** This work was funded by the National Institutes of Health to the Dartmouth Cystic Fibrosis Training Program (T32 HL134598 to ADR) and the Dartmouth Cystic Fibrosis Research Center (DartCF) (P30 DK117469 to JBB), as well as STANTO19R0 (to JBB) from the Cystic Fibrosis Foundation, and the Philip Hanlon and Gail Gentes Cluster for Personalized Treatments for Cystic Fibrosis (to JBB). The funders had no role in study design, data collection and analysis, decision to publish, or preparation of the manuscript.

**Competing interests:** The authors have declared that no competing interests exist.

## Author summary

*P. aeruginosa* is responsible for most airway infections in people with CF. During infection *P. aeruginosa* can use a T3SS to secrete the ADPRT effector ExoS into neutrophils. The T3SS and ExoS each have impacts on bactericidal and inflammatory cell death responses in neutrophils. The majority of *P. aeruginosa* strains isolated from people with CF carry the ExoS gene, suggesting the effector helps establish infection, yet these strains typically have lost T3SS function over time, indicating ExoS is dispensable for chronic infection. However, we and others have identified *P. aeruginosa* isolates from chronic CF infections that have wildtype or hyperactive T3SS function and ExoS secretion, and the hypervirulent strains have been associated with severe lung disease. Here we developed an *ex vivo* infection model to better understand how ExoS+ *P. aeruginosa* strains with wildtype or hyperactive T3SS function impact neutrophil immune responses. We discovered that ExoS ADPRT activity promotes neutrophil inflammatory death that is independent from the response to the T3SS, and results in enhanced PMR with the hyperactive T3SS mutant. These results provide new insights into the mechanism of ExoS as a *P. aeruginosa* virulence factor in CF for infection establishment by wildtype strains, and chronic disease by hypervirulent isolates.

## Introduction

*Pseudomonas aeruginosa* is a Gram negative opportunistic bacterial pathogen responsible for a large percentage of airway infections that cause high morbidity and mortality in people with immune deficiencies, such as cystic fibrosis (CF) [1,2]. *P. aeruginosa* has many virulence factors including a type III secretion system (T3SS) that delivers effectors into host cells [3,4]. Three T3SS effectors, ExoT, ExoU and ExoS are important virulence factors [4,5]. All *P. aeruginosa* strains have ExoT, and ExoU or ExoS, but rarely both [6].

ExoS is a bifunctional protein with a GAP domain and an ADP ribosyltransferase (ADPRT) domain [4]. The GAP domain targets Rho GTPases, whereas the ADPRT domain modifies multiple proteins, such as Ras [4]. The ADPRT domain of ExoS plays a major role in *P. aeruginosa* virulence during infection by blocking phagocytosis and the NADPH oxidase in neutrophils [7–9]. ExoS ADPRT activity is important for the PAO1 laboratory strain to survive in neutrophils [8,10]. ExoT is similar to ExoS in structure and also inhibits the NADPH oxidase in neutrophils [8,10] but is less important for *P. aeruginosa* virulence in airway infections [7,9].

Infection of neutrophils with T3SS+ bacterial pathogens can result in activation of caspase-1 inflammasomes and secretion of the cytokine IL-1β [11]. The inflammasome sensor NLRP3 requires the ASC adaptor to interact with caspase-1, while NLRC4 uses a NAIP as a sensor and can interact with caspase-1 directly or through ASC [11]. Activation of the NLRC4/caspase-1 inflammasome in neutrophils in response to PAO1 infection and T3SS needle and rod, as well as T3SS-mediated secretion of flagellin results in IL-1β secretion [12,13]. Recent studies with PAO1 show that NLRC4/caspase-1 inflammasome activation in neutrophils infected with *P. aeruginosa* can also result in cell death by pyroptosis, which requires pore formation by gasdermin-D (GSDMD) [14–16]. In addition, and somewhat paradoxically, ExoS dampens NLRC4 inflammasome activation [15,16] and drives IL-1β secretion via NLRP3 [16] in neutrophils infected with PAO1. ExoS in PAO1 can also cause damage to neutrophil plasma membranes, as shown by uptake of Sytox Green and release of LDH, and in both cases this activity was independent of caspase-1 [15]. Furthermore, neutrophils infected *in vitro* with

*exoS*[+] strains like PAO1 exhibit signs of NETosis, including citrullination of histone 3 (CitH3), decondensation of nuclear DNA, breakdown of the nuclear membrane, expansion of nuclear DNA into the cytosol, and occasionally extrusion of neutrophil extracellular traps (NETs) [15–17]. Santoni et al. reported that most neutrophils infected with PAO1 fail to extrude NETs (incomplete NET extrusion), possibly due to maintenance of the cortical actin cytoskeleton, a process referred to as "incomplete NETosis" [15].

CF is an inherited disease caused by mutations in the CFTR gene which alter movement of chloride and other ions across cell membranes [18]. This produces highly viscous mucus in lungs which creates a favorable environment for opportunistic bacterial infection. Mortality in people with CF is often due to respiratory failure from chronic bacterial infection. ExoS[+] *P. aeruginosa* strains cause most CF infections [6,19]. The basis for the prevalence of *exoS*[+] *P. aeruginosa* strains in CF airway infections is unknown and remains an important knowledge gap. *P. aeruginosa* undergoes genetic diversification to generate clonally related populations of variant bacteria during chronic CF airway infections [20]. The T3SS may be important for initial colonization of the CF airway, though it is typically inactivated in *P. aeruginosa* during chronic infections [1,21,22]. However, T3SS function is maintained or hyperactive in some *P. aeruginosa* CF clinical isolates, and this phenotype can be associated with hypervirulence and lung function decline [23,24]. Hyperactive T3SS *P. aeruginosa* strains identified to date contain codon changes *exsD*, which is a negative regulator of the T3SS [23,24]. Two *exoS*[+] CF isolates with hyperactive T3SS phenotypes have codon change mutations T188P or S164P in *exsD* [23,24]. Amplicon sequencing of CF isolates showed that the frequency of *P. aeruginosa* with the S164P hyperactive *exsD* allele increased during lung function decline, eventually resulting in terminal respiratory failure [23]. ExsD binds to and sequesters the T3SS transcription factor ExsA [25]. The T188P and S164P codon changes in *exsD* are thought to relieve this negative regulation mechanism, resulting in hypersecretion of ExoS and ExoT, increased *P. aeruginosa* survival in neutrophils and hypervirulence [23,24]. The outcome of interaction of hyperactive T3SS mutants of *P. aeruginosa* with neutrophils in terms of ExoS-promoted inflammasome modulation, pyroptosis or NETosis has not been reported.

In this study we investigated the outcome of interaction of *P. aeruginosa* with neutrophils in terms of ExoS-promoted inflammasome modulation, pyroptosis and NETosis using PAO1 and CF clinical strains including hyperactive T3SS mutants of *P. aeruginosa*. We studied neutrophils from C57BL/6 mice or isogenic animals engineered to have the most common CFTR mutation (F508del) in the human population, or defects in inflammasome assembly (Asc[-/-] or Casp1[-/-]). Our results confirm that ExoS activity in PAO1 and clinical strains suppresses the NLRC4 inflammasome [15,16]. In addition, our data indicate that ExoS ADPRT activity promotes plasma membrane rupture (PMR), and this activity is enhanced with the hyperactive T3SS mutant in C57BL/6 and F508del neutrophils. PMR is inhibited by glycine supplementation, suggesting that ninjurin 1 (NINJ1) [26] is required for this process. In addition, ExoS-promoted PMR was independent of the ASC adaptor and caspase-1, suggesting an inflammasome-independent process. Finally, we find that although ExoS ADPRT promotes generation of CitH3 and PMR, most neutrophils exhibit incomplete NET extrusion, even after infection with the hyperactive T3SS mutant of *P. aeruginosa*. The importance of these findings for understanding the role of ExoS in the pathogenesis of hypervirulent *P. aeruginosa* during neutrophil interactions is discussed.

## Results

### Characterization by flow cytometry of BMNs primed with LPS

For *ex vivo* infections purified bone marrow neutrophils (BMNs) from C57BL/6 (B6) mice were primed with LPS for ~18 hr to upregulate inflammasome components [14]. Notably,

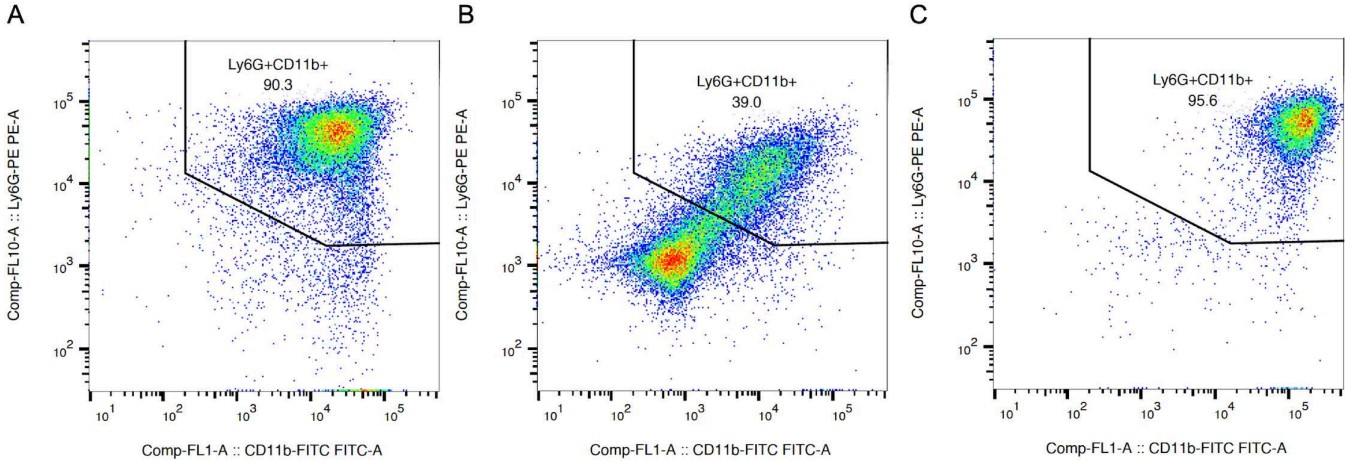

**Fig 1. Characterization of BMNs primed with LPS by flow cytometry.** BMNs were isolated from B6 mice and analyzed immediately (A) or after ~18 hr incubation without (B) or with (C) 100 ng/ml LPS. BMNs were stained with e780, Ly6G-PE and CD11b-FITC and analyzed by flow cytometry. Dead cells (e780+) were excluded from the analysis. Representative dot plots of Ly6G and CD11b signals are shown. The gate indicates live cells that are Ly6G+ and CD11b+.

LPS priming can extend neutrophil life spans *ex vivo* [27,28]. To determine the viability and purity of BMNs, flow cytometry was performed on freshly purified cells as well as purified cells incubated for ~18 hr with or without 100 ng/ml LPS. BMNs were stained with e780 to assess viability and with neutrophil markers Ly6G and CD11b to measure purity (S1 Fig). Dot plots show gating on live cells (e780-), and Ly6G+ and CD11b+ signals (Fig 1). The primed cells (panel C) had a similar percentage of Ly6G+ CD11b+ as the freshly purified BMNs (panel A). In contrast, BMNs incubated without LPS showed a lower percentage of Ly6G+ CD11b+ (panel B). These data indicate that our procedure results in relatively pure populations of primed BMNs for infection experiments.

### ExoS ADPRT activity inhibits IL-1β release in BMNs infected with PAO1

Primed BMNs from B6 mice or those with a Cftr F508del genotype were infected with PAO1-derived strains (Table 1) [10] to confirm that ExoS ADPRT activity is required for *P. aeruginosa* to inhibit NLRC4 inflammasome activation [16]. Results show that more IL-1β was released, as measured by ELISA, from BMNs when they were infected with an ExoS ADPRT catalytic mutant (ExoS(A-)) than the wildtype (WT) control PAO1F (Figs 2A and S2A). As reported by Minns et al. [16] BMNs infected with an ExoT ADPRT catalytic mutant did not release more IL-1β as compared to the control (S2A Fig). PMR, as measured by LDH release, also trended higher with ExoS(A-) but was not significant compared to PAO1F (Figs 2B and S2B). PMR in BMNs infected with the ExoT(A-) mutant was similar to PAO1F (S2B Fig), as expected from the IL-1β data (S2A Fig). Additionally, results from CFU assays indicate that the ability of PAO1F to inhibit IL-1β release compared to ExoS(A-) was not due to increased survival of the WT bacteria during the 60 min infection (Fig 2C). ExoT functions redundantly with ExoS to inhibit the neutrophil NADPH oxidase [8], explaining why ExoS(A-) was not more sensitive to killing than PAO1F (Fig 2C). When comparing BMNs from B6 vs. F508del mice there were no major differences between the two in responses to infection with PAO1F or the ExoS(A-) mutant (Fig 2). Together, these data confirm that the ADPRT activity of ExoS but not ExoT is required for PAO1F to inhibit NLRC4 inflammasome activation and IL-1β release in BMNs [16]. In addition, the

**Table 1. *P. aeruginosa* strains used in this study.**

| Strain | Relevant genotype or description | Reference or Source |
|---|---|---|
| PAO1F | Wild type PAO1 | [10] |
| Δ*pscD* | PAO1F Δ*pscD* | [10] |
| ExoS(A-) | PAO1F *exoS*(ADPRT-) | [10] |
| ExoT(A-) | PAO1F *exoT*(ADPRT-) | [10] |
| ExoS(A-) ExoT(A-) | PAO1F *exoS*(ADPRT-) *exoT*(ADPRT-) | [10] |
| p32_08 | CF isolate patient 32 #8 | [29] |
| p32_85 | CF isolate patient 32 #85 *exsA*$^{T48I}$ | [29] |
| p32_86 | CF isolate patient 32 #86 *exsA*$^{T48I}$ | [29] |
| p32_108 | CF isolate patient 32 #108 | [29] |
| p32_08 ExsA$^{T48I}$ | CF isolate patient 32 #8 *exsA*$^{T48I}$ | This study |
| p32_85 ExsA$^{WT}$ | CF isolate patient 32 #85 *exsA*$^{I48T}$ | This study |
| p32_08 ExoS(A-) | CF isolate patient 32 #8 *exoS*(ADPRT-) | This study |
| p32_85 ExoS(A-) | CF isolate patient 32 #85 *exoS*(ADPRT-) | This study |
| LU ExsD$^{T188P}$ | CF Isolate from upper lobe of left lung from patient 1 | [24] |
| LL ExsD$^{WT}$ | CF Isolate from lower lobe of left lung from patient 1 | [24] |

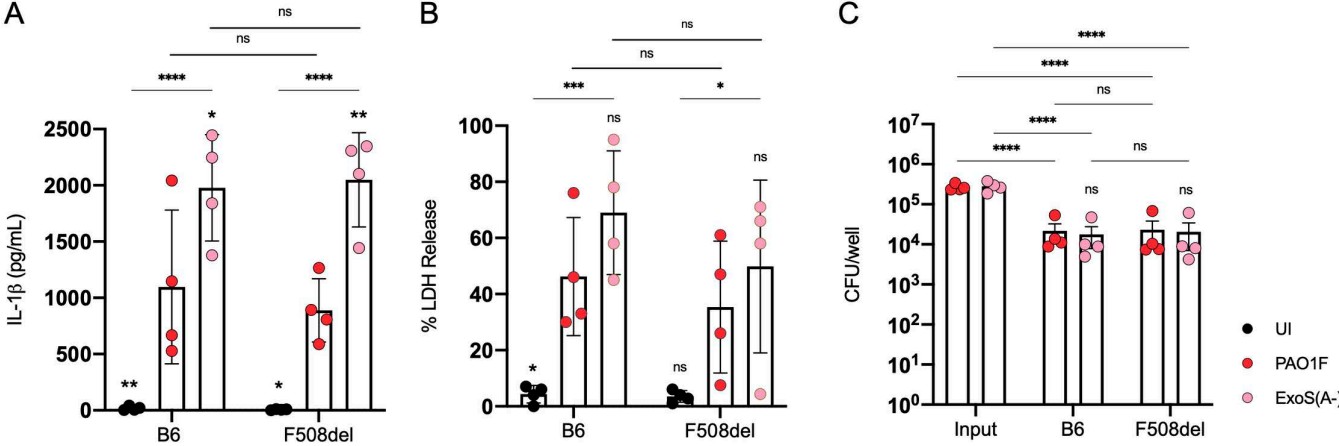

**Fig 2. Analysis of BMN infections with PAO1F or ExoS(A-).** B6 or F508del BMNs were left uninfected (UI) or infected for 60 min with PAO1F or ExoS(A-) at MOI 10 and analyzed for released IL-1β (A) or LDH (B). (C) BMNs were infected as above at an MOI of 1. Samples of bacteria inoculated into each well (Input, T=0 min) and total bacterial-neutrophil co-cultures solubilized in 0.1% NP-40 detergent (T=60 min) were processed by serial dilution and plating to determine CFUs. Data represent normalized values for 2.5x10$^5$ cells/well ± the standard deviation (A, B) or standard error of mean (C) from four independent experiments (A, B, C). Significant differences were determined by two-way ANOVA, comparing to PAO1F within groups or comparing between conditions as shown by brackets. ns, not significant; * P<0.05; ** P<0.01; *** P<0.001; **** P<0.0001.

Cftr genotype of the BMNs did not lead to substantial changes in IL-1β release or bactericidal activities.

## Identification and characterization of hyperactive T3SS mutant *P. aeruginosa* CF clinical isolates

To determine if the use of clinical isolates would impact our results, we studied four clonal isolates that were identified as *exoS*$^+$ *P. aeruginosa* strains that remained T3SS$^+$ over time

in the sinuses of an individual with CF (patient 32) (Table 1) [29]. Sinus infections are common in people with CF [29,30], and they may act as reservoirs that seed *P. aeruginosa* into the lower respiratory tract. Interestingly, genomic sequencing of these strains showed that two of these isolates, p32_85 and p32_86, contain a codon change in *exsA* [29]. Although initially reported as a T68I codon change [29], through additional analysis we determined that the mutation in *exsA* in these isolates results in T48I. We analyzed these four isolates for the ability to secrete ExoS *in vitro* [10]. Using PAO1F and a T3SS null Δ*pscD* mutant (Table 1) as controls, we found that all strains except Δ*pscD* secreted ExoS, and ExoS was hypersecreted by p32_85 and p32_86 (S3A Fig). The previously reported CF isolate with a hyperactive T3SS phenotype due to codon change T188P in *exsD* [24], also hypersecreted ExoS as compared to the control strain (S3B Fig). These results confirm that patient 32 isolates are T3SS⁺ [29] and suggest that the *exsA*^T48I codon change in p32_85 and p32_86 is a gain of function mutation leading to the hyperactive T3SS phenotype. B6 BMNs were infected with the four clinical isolates or PAO1F or Δ*pscD* mutant controls and IL-1β release and PMR were measured. As shown in S4A Fig, BMNs infected with the four clinical isolates released similar amounts of IL-1β. Interestingly, the hyperactive T3SS mutant p32_85 caused enhanced PMR in BMNs as compared to the isolates without the *exsA*^T48I codon change (p32_08 and p32_108, S4B Fig). The hyperactive T3SS mutant p32_86 showed the same trend (S4B Fig). To determine if the codon change in *exsA* was sufficient for the ExoS hypersecretion phenotype, allelic exchange was carried out on isolates p32_08 and p32_85 to mutate codon 48. A T48I codon change was introduced into p32_08, and the mutation in p32_85 was corrected to the WT sequence (Table 1). S3C Fig demonstrates that the codon changes in *exsA* reversed the ExoS secretion phenotype in each clinical strain, indicating that the T48I codon change is sufficient to cause the hyperactive T3SS in isolate p32_85. B6 BMNs were infected with the p32_08 or p32_85 ExsA "allele swap" strains or parent controls and IL-1β release and PMR were measured. BMNs infected with the allele swap strains or the parent controls released similar amounts of IL-1β (S5A Fig). Swapping *exsA*^T48I to the WT allele in p32_85 reversed the enhanced PMR phenotype in BMNs (S5B Fig). Similarly, there was a trend toward enhanced PMR in BMNs when the WT *exsA* allele in p32_08 was swapped to *exsA*^T48I (S5B Fig). These results indicate that the T48I codon change in p32_85 is sufficient to cause enhanced PMR in infected BMNs.

T48I is in the N-terminal regulatory domain of ExsA that interacts with ExsD [25,31–33], suggesting that this mutation relieves negative regulation of the T3SS by reducing binding of ExsA to ExsD. To examine this possibility, we first used AlphaFold2 to predict the structure of the ExsA-ExsD complex, and we highlighted the locations of the T48I mutation, as well as the previously identified hyperactivating S164P and T188P codon changes in ExsD [23,24] (S6A Fig). T48 is a solvent exposed residue on a strand of the beta barrel regulatory domain [32] and as shown in S6A Fig, is located at the predicted interface of the two proteins, raising the possibility that the T48I change (polar to hydrophobic) interferes with the interaction.

A bacterial two hybrid (B2H) assay [32] was used to determine if the T48I codon change impacts ExsA-ExsD interaction. As shown in S6B Fig, the ExsA^T48I protein showed a slight trend toward reduced interaction with ExsD, but this difference was not significant compared to the ExsA control. Shrestha et al. determined that Y24P and V26P codon change variants of ExsA interacted with ExsD by B2H, however were relieved of ExsD inhibition in an in vitro transcription assay [32]. These results suggest that additional experiments will be needed to determine if decreased interaction with ExsD is a major reason for the hyperactive phenotype of the ExsA^T48I protein.

## Hyperactive T3SS mutant *P. aeruginosa* promotes enhanced PMR and reduced bacterial killing in BMNs

We next compared the outcome of infection of BMNs from B6 or F508del mice with CF isolates p32_08 and p32_85. As shown in Fig 3, levels of IL-1β released were low and similar when B6 BMNs were infected with p32_08 and p32_85 (panel A), but p32_85 stimulated higher LDH release, indicating enhanced PMR, compared to p32_08 (panel B). In addition, the clinical isolates p32_08 and p32_85 appeared to survive better in BMNs overall as compared to PAO1F (compare Fig 3C with Fig 2C). p32_85 showed a trend toward enhanced survival compared to p32_08, however it was not a significant viability advantage (Fig 3C). As was seen in Fig 2 there was no substantial difference when comparing between the B6 and F508del BMNs in their responses to infection (Fig 3).

## Hyperactive T3SS mutant *P. aeruginosa* promotes enhanced CitH3 formation in BMNs

To investigate if ExoS activity and the hyperactive T3SS phenotype impacts NETosis processes, we analyzed levels of CitH3 in *P. aeruginosa*-infected BMNs. B6 and F508del BMNs were infected with PAO1F, ExoS(A-), p32_08 or p32_85 and immunoblotting was used to measure generation of CitH3 by the calcium regulated enzyme PAD4 [14]. Cleavage of GSDMD by caspase-1 was measured as a control since ExoS ADPRT activity inhibits this process [16]. As shown in Fig 4A, GSDMD cleavage was highest in B6 BMNs infected with ExoS(A-) as expected [16], although some processing was also detected with PAO1F, p32_08, and p32_85, indicating the ExoS inhibition is not complete. As compared to uninfected, increased CitH3 was generated in response to BMN infection with PAO1F and ExoS(A-), and the elevated production with ExoS(A-) was expected, since NLRC4 activation is associated with this activity [14,15] (Fig 4A). Interestingly, CitH3 was also elevated with p32_85 compared to p32_08 (Fig 4A), suggesting ExoS ADPRT activity in the T3SS hyperactive mutant is driving this process independent of NLRC4 activation. Similar results overall were obtained with F508del

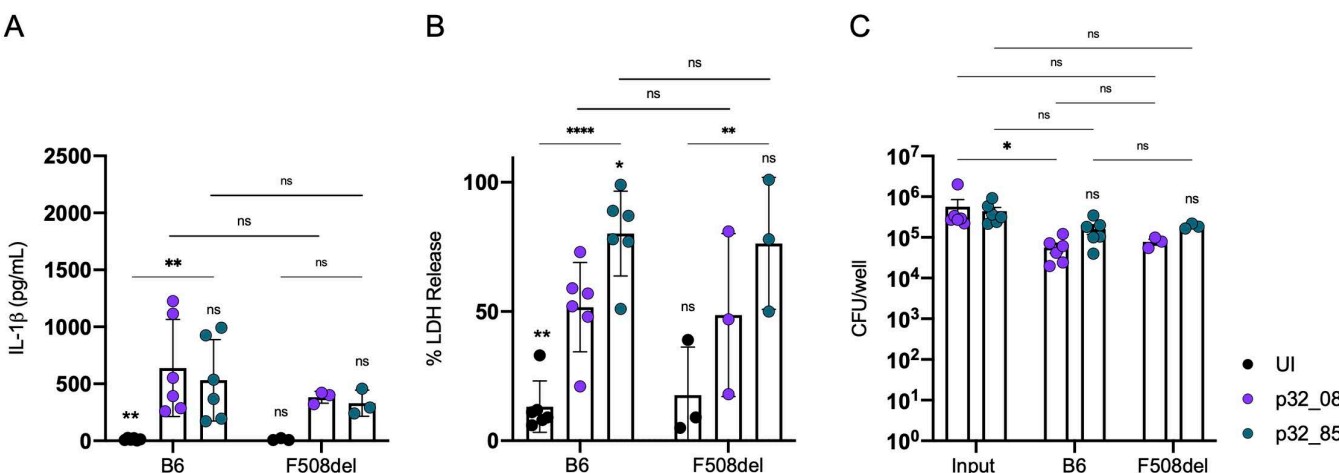

**Fig 3. Analysis of BMN infections with patient 32 isolates.** B6 or F508del BMNs were left UI or infected for 60 min with p32_08 or p32_85 at MOI 10 (A, B) or 1 (C) and analyzed for released IL-1β (A) or LDH (B) or CFU (C). Data represent normalized values for $2.5 \times 10^5$ cells/well ± the standard deviation (A, B) or standard error of mean (C) from 6 (B6) or 3 (F508del) independent experiments (A, B, C). Significant differences were determined by two-way ANOVA comparing to p32_08 within groups or comparing between conditions as shown by brackets. ns, not significant; * P<0.05; ** P<0.01; **** P<0.0001.

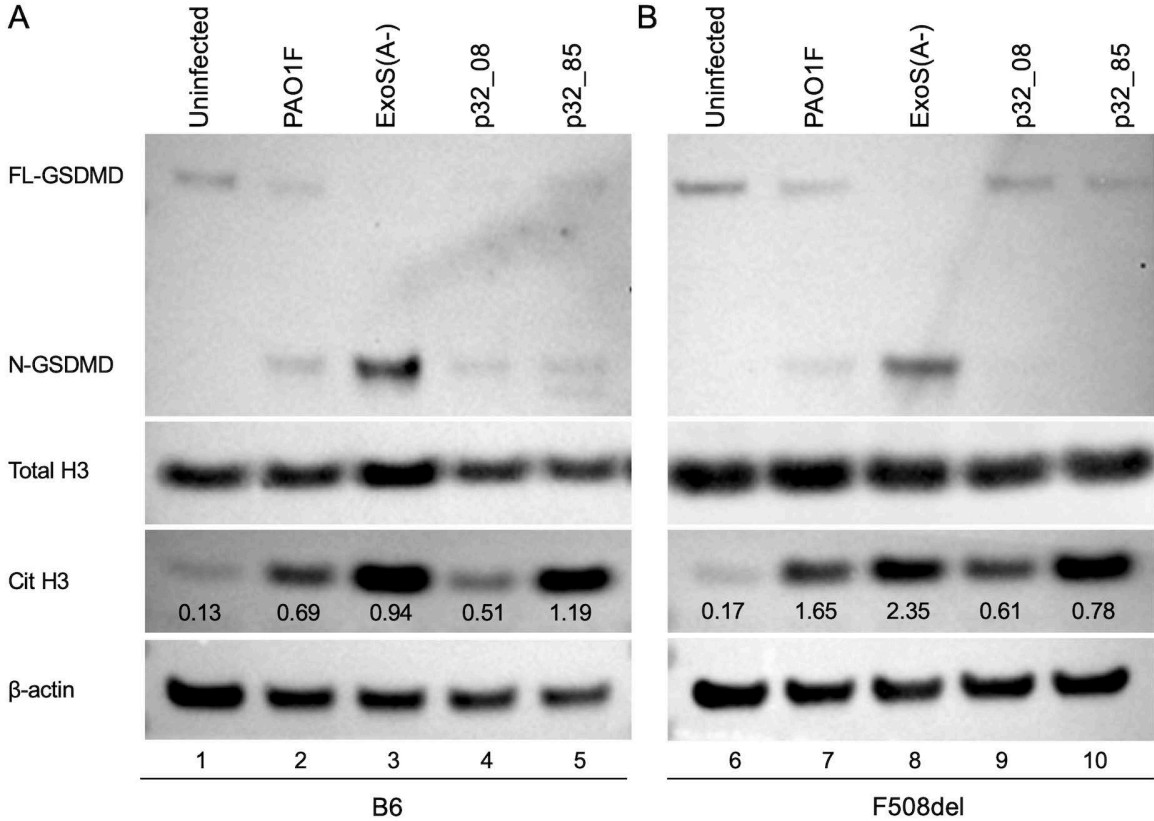

**Fig 4. Analysis of BMN infections with PAO1F strains or patient 32 isolates by immunoblotting.** BMNs from B6 (A) or F508del (B) mice were left uninfected or infected with the indicated strains of PAO1F or patient 32 isolates for 60 min at an MOI of 10. Samples of total well contents were analyzed by immunoblotting for full length (FL-) or cleaved (N-) GSDMD, total histone 3 (H3), citrulli-nated histone 3 (Cit H3), and β-actin as a loading control. One representative blot of three independent experiments is shown. Values listed under Cit H3 represent average signal intensity of the ratio of Cit H3 over total H3 for three independent experiments.

BMNs (Fig 4B), indicating that ExoS ADPRT activity is promoting enhanced CitH3 formation in both CF and non-CF neutrophils infected with the hyperactive T3SS mutant.

## Hyperactive T3SS mutant *P. aeruginosa* promotes nuclear DNA decondensation and incomplete NET extrusion in BMNs

Since CitH3 is a marker for chromatin decondensation, and a prerequisite for NET extrusion, we used live cell microscopy to determine if infection with PAO1F or our clinical isolates was promoting these outcomes in BMNs. We first compared infections with PAO1F and PA14 at 60, 120 and 180 min, as the latter ExoU+ strain was expected to serve as a positive control in microscopy to visualize chromatin decondensation and NET extrusion in B6 BMNs [15]. Decondensed chromatin and extruded NETs were detected with the membrane-impermeant DNA dye Sytox Green, and Hoechst dye was used to label DNA in BMNs with intact mem-branes. The percentage of cells positive for Sytox was determined by automated analysis and BMNs with extruded NETs were counted visually. We observed with PA14 at 60 min, in comparison to uninfected, that there was significantly increased Sytox positivity and a trend toward more extruded NETs in infected BMNs (Fig 5A, 5D, and 5E). NET extrusion with PA14 increased at 120 and 180 min to an extent that quantification of Sytox or NET positivity on a per cell basis was deemed inaccurate and not undertaken (Fig 5B-E). In BMNs infected

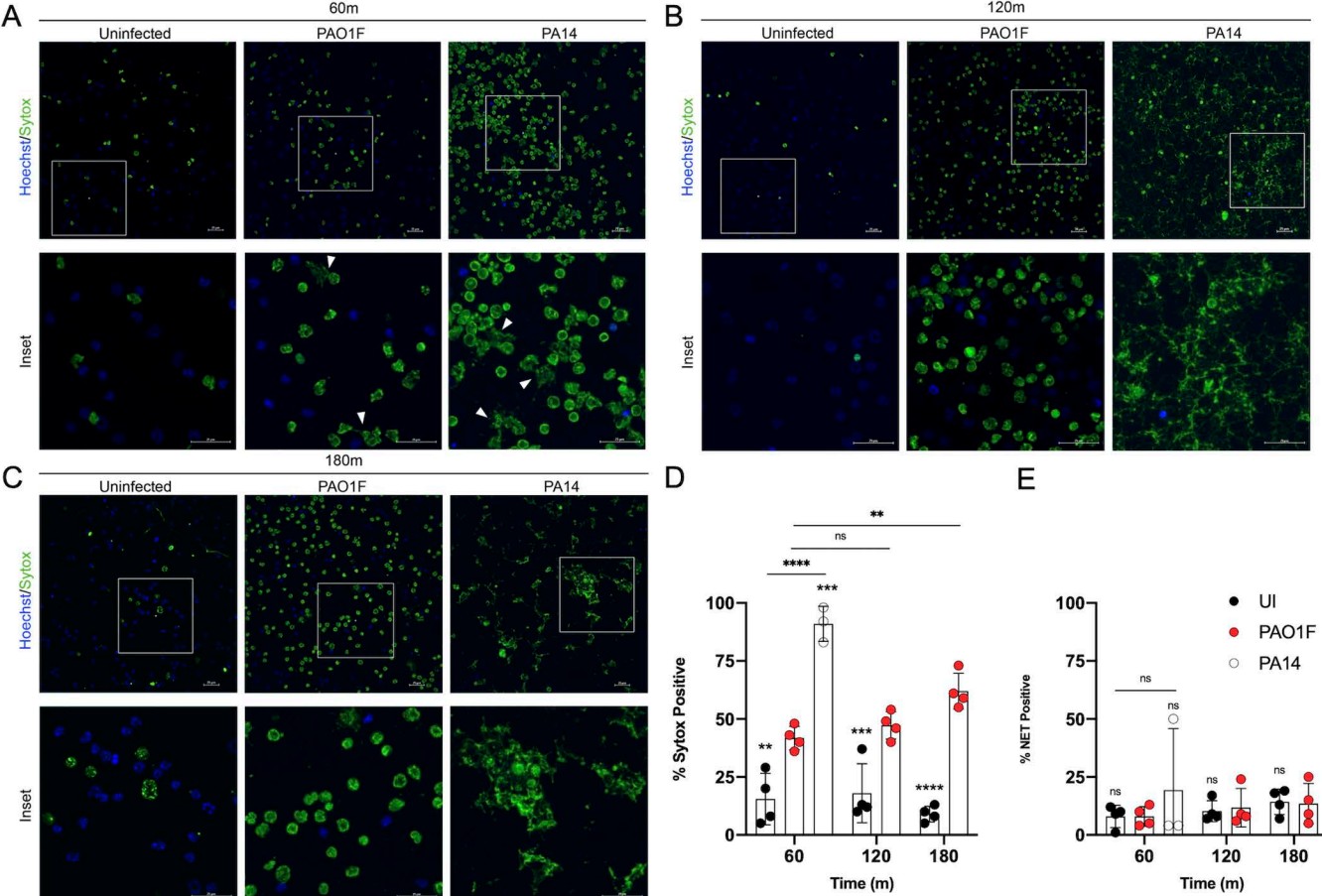

**Fig 5. Live cell imaging of BMNs infected with PAO1F or PA14 laboratory strains.** B6 BMNs were left uninfected/UI or infected for 60 (A), 120 (B), or 180 (C) min with the indicated strains at MOI 10. Cells were stained with Hoechst and Sytox Green 30 min prior to imaging with a spinning disc confocal microscope. Top images are representative of entire wells taken at 40x. Scale bars are 25 μM. Insets show areas magnified in lower images. In (A) BMNs with NETs are shown with white arrow heads. (D) Percent of total BMNs that were Sytox Green positive from three or four independent experiments as determined by automated cell counting of images. Values for PA14 at 120 and 180m were not determined. (E) Percent of Sytox Green positive cells with NETs from three or four independent experiments as determined by visual counting. Significant differences were determined by two-way ANOVA comparing to PAO1F at each time point or between conditions as shown by brackets. ns, not significant; ** P<0.05; *** P<0.01; **** P<0.0001.

for 60 min with PAO1F Sytox positivity was enhanced compared to uninfected but less than PA14 (Fig 5A and 5D). The percent of BMNs infected with PAO1F that were Sytox positive increased over time but NET extrusion remained at baseline (Fig 5B-E). We next imaged BMNs infected with PAO1F, ExoS(A-), p32_08 or p32_85. The increased CitH3 seen in BMNs infected with ExoS(A-) and the hyperactive T3SS mutant p32_85 as compared to controls (Fig 4) was correlated with greater Sytox positivity but not enhanced NET extrusion (Fig 6A-E). These results indicate that hyperactive T3SS mutant *P. aeruginosa* drives incomplete NET extrusion in BMNs, like the NLRC4-mediated outcome in neutrophils infected with ExoS(A-) (Fig 6) or an *exoU* mutant of *P. aeruginosa* [15].

## ASC contributes to IL-1β release but is dispensable for enhanced PMR in BMNs infected with hyperactive T3SS mutant *P. aeruginosa*

When evaluating potential mechanisms for the enhanced PMR in BMNs infected with hyperactive T3SS mutant *P. aeruginosa* we considered the possible role of inflammasome activation.

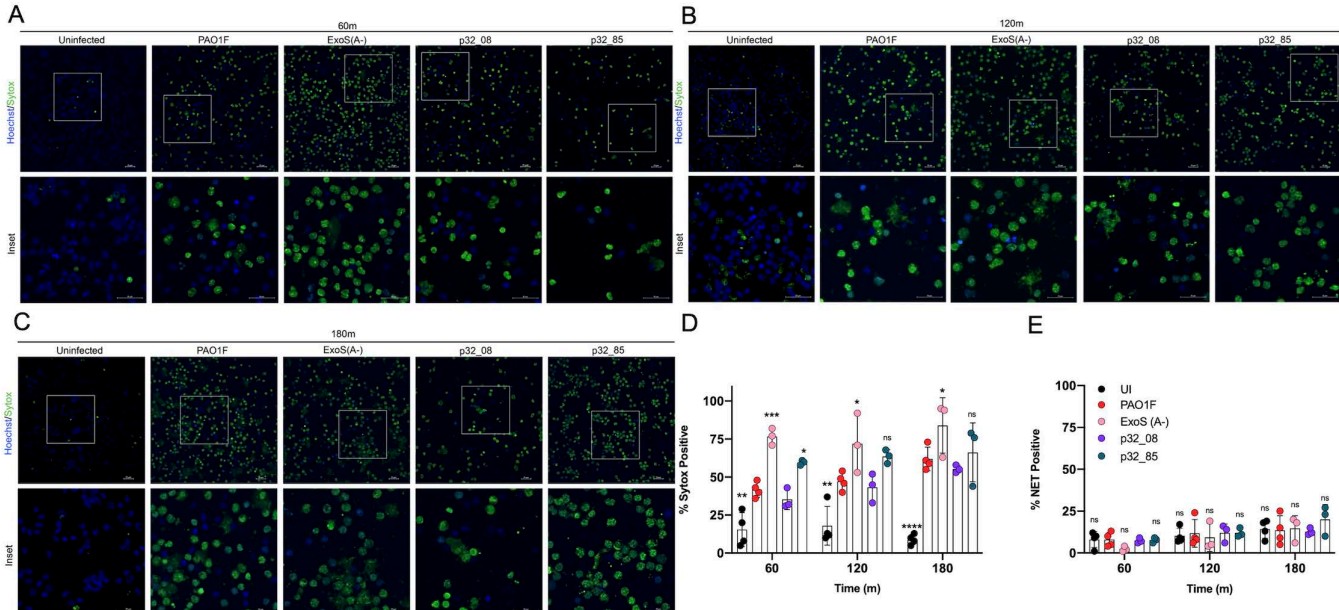

**Fig 6. Live Cell Imaging of BMNs infected with PAO1F or patient 32 isolates.** B6 BMNs were left uninfected/UI or infected with indicated strains and processed and analyzed as described in the legend to Fig 5. Significant differences were determined by two-way ANOVA comparing to PAO1F for UI or ExoS (A-) or comparing to p32_08 for p32_85 at each time point. ns, not significant; * P<0.05; ** P<0.01; *** P<0.001; **** P<0.0001.

Although ExoS suppresses the NLRC4 inflammasome [15,16], release of IL-1β and cleavage of GSDMD is not completely blocked (Figs 2, S2, 3 and 4), and Minns et al. obtained evidence that ExoS ADPRT activity drives activation of NLRP3 in BMNs infected with PAO1 [16]. To examine the role of inflammasome activation in PMR we first used Asc[-/-] BMNs, which lack the adaptor that is required for, or can contribute to, NLRP3 and NLRC4 inflammasome assembly, respectively [34]. B6 or Asc[-/-] BMNs were infected with PAO1F, ExoS(A-), p32_08 or p32_85 and LDH release was determined to measure PMR. We also measured secreted IL-1β and immunoblotted for GSDMD cleavage and CitH3. Fig 7B shows that PMR was not significantly reduced for any of the strains comparing B6 to Asc[-/-] BMNs. Strain p32_85 continued to cause a trend toward enhanced PMR in Asc[-/-] BMNs as compared to p32_08 (Fig 7B). Notably, IL-1β release was significantly reduced for all strains in the absence of ASC (Fig 7A). There were corresponding decreases in cleavage of GSDMD in the absence of ASC for all strains (Fig 7C). These results indicate a possible role for ASC in the NLRC4 inflammasome triggered by ExoS(A-) infection, and the NLRP3 inflammasome driven by ExoS ADPRT activity [16] with the other strains (Fig 7A). Interestingly, CitH3 production was reduced in the absence of ASC for ExoS(A-) but not p32_85 (Fig 7C). Together, these finding suggests that ExoS ADPRT-promoted PMR and CitH3 production in response to infection with the hyperactive T3SS mutant is ASC-independent.

## Caspase-1 contributes to CitH3 but is dispensable for enhanced PMR in BMNs infected with hyperactive T3SS mutant *P. aeruginosa*

Casp1[-/-] cells were used to evaluate the importance of Caspase-1 for the enhanced PMR in BMNs infected with hyperactive T3SS mutant *P. aeruginosa*. BMNs were infected with the clinical strains p32_08 or p32_85, as well as their corresponding ADPRT catalytic activity mutants (ExoS(A-)) constructed by allelic exchange (Table 1). LDH release was determined to measure PMR. Secreted IL-1β was measured by ELISA, as well as GSDMD cleavage and

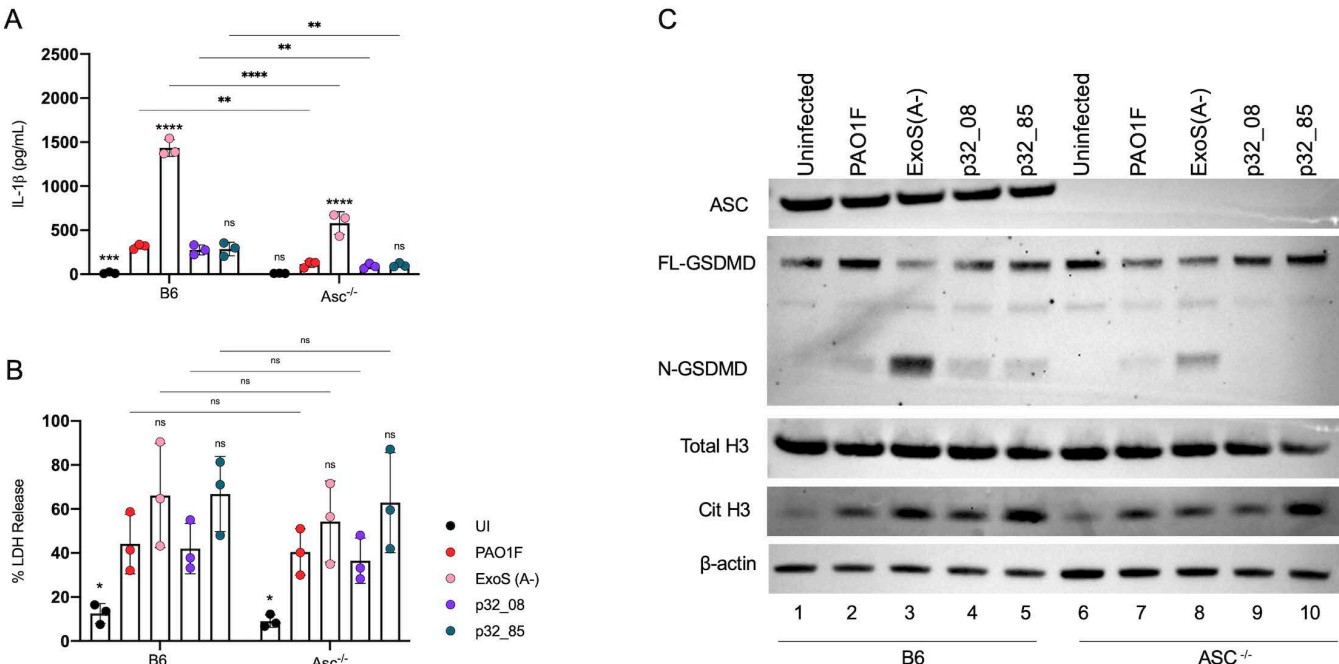

**Fig 7. Analysis of B6 or ASC[-/-] BMN infections with PAO1F strains or patient 32 isolates.** B6 or ASC-/- BMNs were left uninfected/UI or infected for 60 min with indicated strains at MOI 10 and analyzed for released IL-1β (A) or LDH (B). Data represent normalized values for 2.5x10⁵ cells/well ± the standard deviation from three independent experiments. (C) Samples of total well contents were analyzed by immunoblotting for ASC, full length (FL-) or cleaved (N-) GSDMD, total histone 3 (H3), citrullinated histone 3 (Cit H3), and β-actin as a loading control. One representative blot of three independent experiments is shown. (A,B) Significant differences were determined by two-way ANOVA comparing PAO1F to UI and ExoS (A-) or comparing p32_08 to p32_82 within groups or comparing between conditions as shown by brackets. ns, not significant; * $P<0.05$; ** $P<0.01$; *** $P<0.001$; **** $P<0.0001$.

CitH3 by immunoblotting. The use of the ExoS(A-) mutants confirmed that ADPRT activity is required for ExoS in the patient 32 strains to inhibit the NLRC4 inflammasome in B6 BMNs (Fig 8A and 8C). IL-1β release was significantly reduced (Fig 8A) and GSDMD cleavage was diminished (Fig 8C) with the ExoS(A-) mutants in the absence of caspase-1. However, PMR was not significantly reduced for p32_08 or p32_85 but was diminished for the corresponding ExoS(A-) mutants, comparing B6 to Casp1[-/-] BMNs (Fig 8B). We noted that CitH3 was lower with p32_08 ExoS(A-) as compared to p32_08 in B6 BMNs (Fig 8C), which is the opposite of what was seen with the ExoS(A-) mutant of PAO1F vs. the parental strain (Figs 4 and 7C). However, CitH3 was lower with p32_08 ExoS(A-) and p32-85 ExoS(A-) in the absence of caspase-1 as expected (Fig 8C). Notably, CitH3 production in response to infection with p32_85 was also reduced in the absence of caspase-1 (Fig 8C). These results suggest that caspase-1 contributes to CitH3 but is dispensable for enhanced PMR in BMNs infected with hyperactive T3SS mutant *P. aeruginosa*.

## Glycine inhibits PMR promoted by ExoS ADPRT activity in BMNs infected with hyperactive T3SS mutant *P. aeruginosa*

Recently a new mechanism for PMR has been identified, in which the NINJ1 protein oligomerizes downstream of different cell death pathways, leading to the formation of large lesions in the plasma membrane [35,36]. PMR has been shown to be inhibited by the addition of the amino acid glycine, which is thought to prevent the oligomerization of NINJ1 [26]. Santoni et al. found that glycine selectively inhibited PMR but not IL-1β release in BMNs infection with

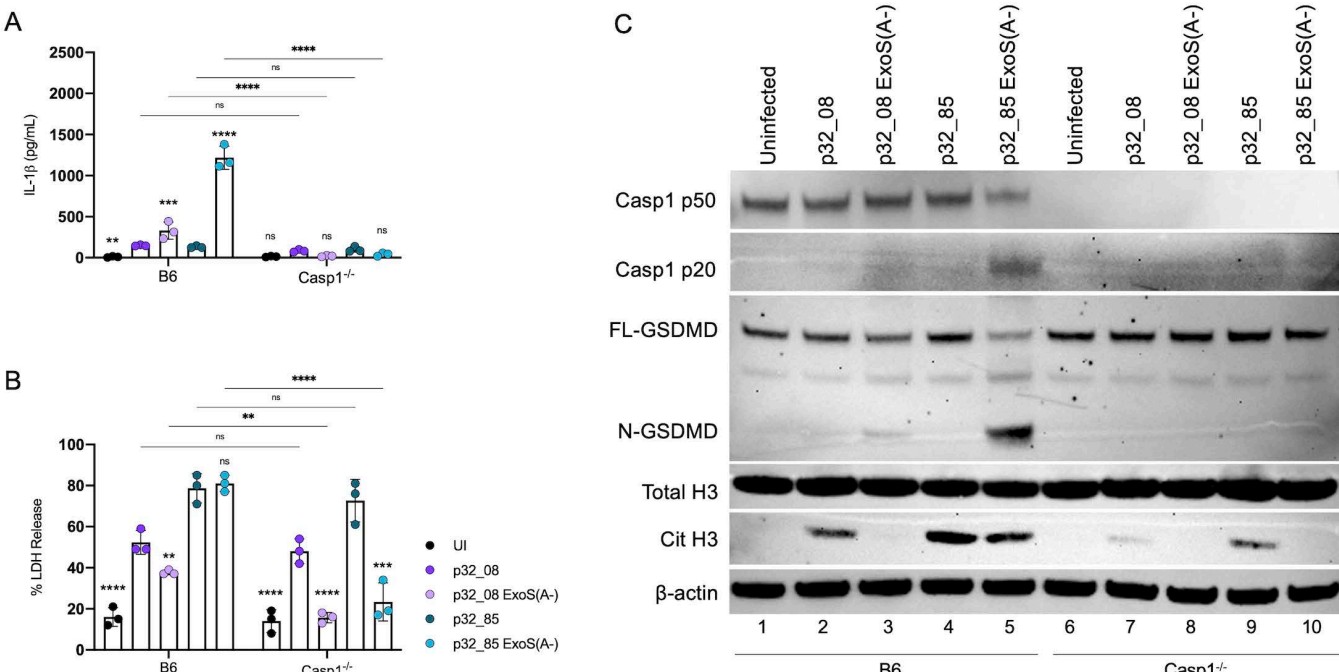

**Fig 8. Analysis of B6 or Casp1[-/-] BMN infections with patient 32 strains.** B6 or Casp1[-/-] BMNs were left uninfected/UI or infected for 60 min with indicated strains at MOI 10 and analyzed for released IL-1β (A) or LDH (B). Data represent normalized values for 2.5x10⁵ cells/well ± the standard deviation from three independent experiments. (C) Samples of total well contents were analyzed by immunoblotting for caspase-1 (Casp1 full length p50 or cleaved p20), full length (FL-) or cleaved (N-) GSDMD, total histone 3 (H3), citrullinated histone 3 (Cit H3), and β-actin as a loading control. One representative blot of three independent experiments is shown. (A, B) Significant differences were determined by two-way ANOVA comparing p32_08 to UI or p32_08(ExsA-) or comparing p32_85 to p32_85 ExsA(A-) within groups or comparing between conditions as shown by brackets. ns, not significant; ** P<0.01; *** P<0.001; **** P<0.0001.

a PAO1 Δ*exoS* mutant, demonstrating a selective cytoprotective effect against NINJ1 lesion formation but not GSDMD pore formation downstream of NLRC4 inflammasome activation [15]. We tested a role for NINJ1 in PMR in our BMN model by adding 5mM glycine to the media during *P. aeruginosa* infection. BMNs were infected with p32_08, p32_08 ExoS(A-), p32_85 or p32_85 ExoS(A-). LDH release was determined to measure PMR, and as controls we also measured secreted IL-1β and immunoblotted for GSDMD cleavage and CitH3. We found that glycine significantly reduced LDH release in all infections, indicating a reduction in NINJ1-mediated PMR (Fig 9B). In contrast, glycine addition had no impact on cleavage of GSDMD (Fig 9C) or release of IL-1β (Fig 9A) in the infected BMNs, as expected. Additionally, we saw no difference in the level of CitH3 between glycine conditions (Fig 9C). Together, these results suggest that NINJ1 can mediate PMR in BMNs in response to either NLRC4 inflammasome activation [15] or ExoS ADPRT activity (Fig 9). In addition, either NLRC4 inflammasome activation or ExoS ADPRT activity can promote formation of CitH3 independent of NINJ1-mediated PMR (Fig 9). In the case of the former, GSDMD pores may promote CitH3 [14] and NINJ1 lesions [15]. The type of membrane perturbation that promotes CitH3 generation and activation of NINJ1 in response to ExoS ADPRT activity is unknown.

## Discussion

Results presented here are consistent with recent reports indicating that upon infection of neutrophils with PAO1 the *P. aeruginosa* T3SS translocates flagellin, which activates the NLRC4 inflammasome, resulting in GSDMD pores, release of IL-1β, pyroptosis, generation

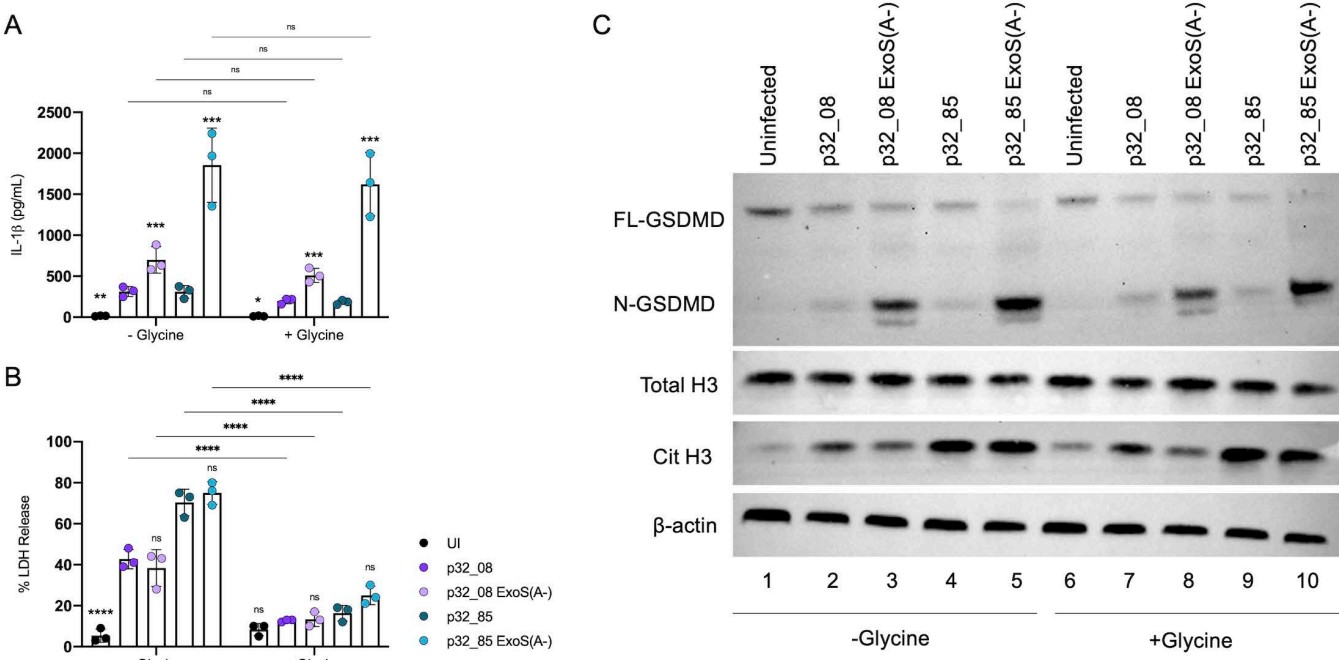

**Fig 9. Analysis of B6 BMNs infections with patient 32 strains in absence or presence of glycine.** B6 BMNs were left uninfected/UI or infected for 60 min with indicated strains at MOI 10 in the absence or presence of 5 mM glycine and analyzed for released IL-1β (A) or LDH (B). Data represent normalized values for 2.5x10^5 cells/well ± the standard deviation from three independent experiments. (C) Samples of total well contents were analyzed by immunoblotting for full length (FL-) or cleaved (N-) GSDMD, total histone 3 (H3), citrullinated histone 3 (Cit H3), and β-actin as a loading control. One representative blot of three independent experiments is shown. (A, B) Significant differences were determined by two-way ANOVA comparing p32_08 to UI or p32_08(ExsA-) or comparing p32_85 to p32_85 ExsA(A-) within groups or comparing between conditions as shown by brackets. ns, not significant; * P<0.05; ** P<0.01; *** P<0.001; **** P<0.0001.

of CitH3 and decondensation of chromatin into incomplete NETs, and activation of NINJ1 leading to PMR [12,14–16]. In addition, our data are in line with the idea that the ADPRT activity of translocated ExoS dampens activation of NLRC4 [15,16]. We identified an ExoS+ *P. aeruginosa* hyperactive T3SS mutant clinical strain from a CF infection with potentially enhanced resistance to neutrophil bactericidal activity, like the hypervirulent isolates identified by Jorth et al. [23,24]. The *P. aeruginosa* hyperactive T3SS mutant was used to obtain new evidence that ExoS ADPRT activity promotes inflammasome-independent CitH3, chromatin decondensation into incomplete NETs, and activation of NINJ1, leading to PMR in CF and non-CF murine neutrophils (S7A Fig). ExoS-promoted PMR results in release of LDH and danger-associated molecular patterns such as HMGB1 are likely to be liberated as well (S7A Fig), with the potential to promote inflammatory signaling. On the other hand, infection of neutrophils with the *P. aeruginosa* hyperactive T3SS mutant translocating catalytically-inactive ExoS, unleashes the flagellin-NLRC4-caspase-1 axis, resulting in GSDMD pores, release of IL-1β, pyroptosis, generation of CitH3 and decondensation of chromatin into incomplete NETs, and activation of NINJ1 leading to PMR (S7B Fig). It is remarkable that two very different *P. aeruginosa* T3SS substrates, ExoS and flagellin, can trigger similar (except for release of IL-1β in the case of the latter) responses in neutrophils (S7 Fig).

The clonal hyperactive T3SS mutants with a T48I codon change in *exsA* were identified by genome sequencing of individual colonies obtained at a single time point (day 553) in a longitudinal study of a person with CF with a chronic *P. aeruginosa* sinus infection [29]. Sinuses are thought to act as reservoirs for generating *P. aeruginosa* mutants with enhanced fitness that

     

seed the lower respiratory tract [29]. Unfortunately, no data is available to suggest increased airway disease in patient 32 at day 553 or if the hyperactive T3SS mutant successfully colonized the lungs [29]. It is also unclear if *exsA*[T48I] was the majority allele in the *P. aeruginosa* population in the sinuses at day 553, or if it was coincidental that the colonies selected contained this mutation. It is possible that patient 32 was fortunate in avoiding dissemination of *exsA*[T48I] from the sinuses, since the two known *exsD* hyperactive T3SS mutants that have been isolated were associated with serious lung function decline in people with CF [23,24]. Going forward it will be important to determine how ExsA[T48I] promotes a hyperactive T3SS phenotype, given that our B2H results suggest that its interaction with ExsD is similar to WT. The ExsD S164P and T188P codon variants also promote a hyperactive T3SS phenotype by an unknown mechanism [23,24], and the proline insertions are likely to disrupt the alpha helices in which they are located (S6A Fig). Defining how these hyperactivating codon changes impact ExsA and ExsD functions will provide additional insight into the interaction mechanism of these two proteins. In vitro transcription assays appear to be a sensitive method to determine how ExsD interaction impacts ExsA function [32]. It will also be interesting to determine if hypervirulent *P. aeruginosa* strains are unknowingly present in strain collections from diverse infections sources, as this would increase the clinical significance of these isolates.

Our experiments prioritize ExoS over ExoT for the following reasons. First, the former is more important for *P. aeruginosa* virulence in airway infections [7,9]; second, the *exoS*[+] gene is epidemiologically linked to successful CF airway infection by *P. aeruginosa* [6,19]; and third ExoS but not ExoT is required to inhibit GSDMD cleavage and IL-1β release from BMNs [15,16] (S2 Fig). In addition, unlike ExoS, ExoT is not required for LDH release in BMNs lacking caspase-1 (Casp1[-/-]) [15], indicating that ExoT does not contribute to PMR. However, ExoT ADPRT activity can independently reduce NADPH oxidase-dependent bactericidal activity against PAO1 in neutrophils [8,10] and the same is likely true for hypervirulent *P. aeruginosa* strains.

The ExoS ADPRT domain has broad specificity for host protein targets making it challenging to determine the mechanism by which it dampens activation of NLRC4 and promotes IL-1β release via the NLRP3 inflammasome, generation of CitH3 and decondensation of chromatin into incomplete NETS, and activation of NINJ1 leading to PMR in murine neutrophils. In addition, it is striking that ExoS does not suppress NLRC4 or activate the NLRP3 inflammasome in murine bone marrow-derived macrophages infected with PAO1 [16]. Minns et al. suggest that ExoS ADPRT inhibits NAIPs/NLRC4 and activates NLRP3 in murine neutrophils by direct modifications, and that increases in the levels of these substrates in murine macrophages makes these cells less sensitive to intoxication [16]. Minns et al. obtained evidence that the priming state of murine neutrophils may also lead to different responses to PAO1 infections due to impacts on levels of inflammasome components and ExoS substrates [16]. It is also possible that ExoS ADPRT activity indirectly inhibits NAIPs/NLRC4 and activates NLRP3, while simultaneously promoting generation of CitH3 and decondensation of chromatin into incomplete NETS, and activation of NINJ1 leading to PMR in neutrophils. An early study reported that ExoS ADPRT activity promotes apoptosis in HeLa cells infected with *P. aeruginosa* [37]. Experiments using inhibitors of capases-8 and -3 could be used to investigate the role of apoptosis pathways in neutrophil responses to ExoS ADPRT activity. However, Minns et al. found no evidence that caspase-8 or -3 were activated in neutrophils infected with PAO1 [16].

Santoni et al. presented preliminary evidence that ExoS contributes to PMR in murine or human neutrophils infected with PAO1 [15], but additional experiments are needed to define if ADPRT activity dampens activation of NLRC4, generation of CitH3 and decondensation of chromatin into incomplete NETS, and activation of NINJ1 leading to PMR in human cells.

     

It will also be important to determine if there is any impact of CF genotype on neutrophil response to ExoS ADPRT activity. Thus far we have not observed major phenotypic differences between non-CF and CF murine BMNs. However, differences between non-CF and CF human neutrophils have been reported in terms of antimicrobial responses and NET extrusion [38] and therefore these comparisons are important.

Pad4 is a calcium-dependent enzyme and evidence suggests that $Ca^{2+}$ influx through GSDMD pores is needed for CitH3 generation in response to NLRC4 inflammasome activation in infected neutrophils [14,15]. Our results suggest that enhanced CitH3 generation in response to infection with hyperactive T3SS mutant *P. aeruginosa* can occur in the absence of detectable GSDMD cleavage in $Asc^{-/-}$ or $Casp1^{-/-}$ neutrophils. ExoS ADPRT activity in neutrophils infected with hyperactive T3SS mutant *P. aeruginosa* may be promoting $Ca^{2+}$ influx by some other mechanism, leading to enhanced CitH3 generation. In S7A Fig we propose that ExoS ADPRT activity leads to opening of plasma membrane pores allowing for $Ca^{2+}$ influx. One source of such pores is the T3SS translocon itself. However, we favor a host-derived pore (S7A Fig) based on evidence that ectopic expression of the active ExoS ADPRT domain in a cell line is sufficient to cause plasma membrane damage [39]. Alternatively, it is possible that GSDMD pore formation below our limit of detection allows $Ca^{2+}$ influx under our infection conditions. This would explain why CitH3 formation was partially reduced in $Casp1^{-/-}$ BMNs infected with hyperactive T3SS mutant *P. aeruginosa*.

Our results using glycine supplementation during BMN infection suggest that PMR promoted by ExoS ADPRT activity is dependent on activation and oligomerization of NINJ1, resulting in large lesions in the plasma membrane [26,35,36] (S7A Fig). How NINJ1 is activated in response to membrane perturbations is unknown. Formation of GSDMD pores appears to be one mechanism leading to NINJ1 activation. Degen et al. suggest that NINJ1 senses changes in membrane composition, such as the exposure of negatively charged phosphatidylserine on the plasma membrane [35]. We considered the possibility that NINJ1-mediated plasma membrane damage was involved in $Ca^{2+}$ influx needed for enhanced CitH3 generation in response to infection with hyperactive T3SS mutant *P. aeruginosa*. However, enhanced CitH3 generation was not reduced by glycine supplementation. This result may indicate that glycine only prevents NINJ1 oligomerization to form large lesions but does not prevent $Ca^{2+}$ influx. Alternatively, ExoS ADPRT activity may promote membrane perturbations that in turn allow $Ca^{2+}$ influx and NINJ1 activation (S7A Fig).

Although the mechanism by which ExoS ADPRT activity promotes generation of CitH3 and decondensation of chromatin into incomplete NETS, and activation of NINJ1 leading to PMR in neutrophils is unknown, these responses could represent cell intrinsic mechanisms of bactericidal or pathogenic activities impacting *P. aeruginosa*. Santoni et al. compared survival of PAO1 in BMNs that were deficient in Pad4 ($Pad4^{-/-}$) or NLRC4 ($Nlrc4^{-/-}$) or control WT [15]. Results indicate that generation of CitH3 and incomplete NET extrusion under control of Pad4 had no impact on bacterial survival, while $Nlrc4^{-/-}$ BMNs were significantly more bactericidal [15]. These results suggest that a neutrophil response under control of NLRC4 is pathogenic. Future experiments will be needed to determine if ExoS-promoted PMR favors survival of *P. aeruginosa* during neutrophil infection. It would be informative to measure survival of hyperactive T3SS mutant *P. aeruginosa* in $Ninj1^{-/-}$ neutrophils to determine if enhanced PMR contributes to the hypervirulent phenotype of these strains.

## Materials and methods

### Ethics statement

Isolation of bone marrow from mice was carried out in accordance with a protocol that adheres to the Guide for the Care and Use of Laboratory Animals of the National Institutes of Health

(NIH) and was reviewed and approved (approval 2148) by the Institutional Animal Care and Use Committee at Dartmouth College. The Dartmouth College animal program is registered with the U.S. Department of Agriculture (USDA) through certificate number 12-R-0001, operates in accordance with Animal Welfare Assurance (NIH/PHS) under assurance number D16-00166 (A3259-01) and is accredited with the Association for Assessment and Accreditation of Laboratory Animal Care International (AAALAC, accreditation number 398).

### Bacterial strains

*P. aeruginosa* strains used in this study are listed in Table 1. *P. aeruginosa* strains for neutrophil infections and secretion assays were grown on Luria-Bertani (LB) agar plates, and in LB high salt broth (11.7g/L NaCl) supplemented with $MgCl_2$ (10mM) and $CaCl_2$ (0.5mM) at 37˚C [10]. *P. aeruginosa* for strain construction were grown in LB broth. *Escherichia coli* for strain construction were grown in LB supplemented with the appropriate antibiotic.

### Plasmids

Primer sequences used to amplify and mutagenize the ExoS and ExsA genes are outlined in S1 Table. The ExoS and ExsA genes were amplified from patient 32 isolates 8 and 85 using standard colony PCR and cloned into pTOPO2.1 and transformed into *E. coli* DH5α. Site directed mutagenesis was performed on pTOPO-ExoS to introduce two point mutations, resulting in E379D and E381D, to inactivate ExoS ADPRT catalytic activity following the protocol provided by Agilent Technologies. All constructs were then cloned from pTOPO into pMQ30 using standard restriction digestion and ligation and transformed into DH5α for long term storage and S17-1 *E. coli* for conjugation with *P. aeruginosa*.

### Conjugation and allelic exchange

Strains of *P. aeruginosa* for conjugation were grown overnight at 42˚C without shaking in LB media. *E. coli* strains carrying the allelic exchange vector were grown overnight at 30˚C with shaking in LB with appropriate antibiotic. Constructs were introduced into *P. aeruginosa* by conjugation with S17-1 *E. coli*. Meridiploids were selected by antibiotic resistance. Double recombinants were acquired using 10% sucrose counter selection [40]. PCR and sanger sequencing was used to confirm mutations.

### Mouse strains

C57BL/6J (Strain #664) and Casp1$^{-/-}$ (Strain #032662) mice were purchased from Jackson Laboratories. Mice with the *Cftr$^{F508del}$* mutation on the C57BL/6 background were obtained from Case Western Reserve University's Cystic Fibrosis Mouse Models Core and bred at Dartmouth [41]. CCR2-GFP$^+$ [42] were bred and housed at Dartmouth college. Asc$^{-/-}$ mice [43] were acquired from the lab of Joshua Obar at Dartmouth College. CCR2-GFP$^+$ BMNs were used as WT controls for Asc$^{-/-}$ BMNs in the experiments shown in Fig 7. All mice used in this study were in the C57BL/6 genetic background and were bred in and housed up to four mice per ventilated cage at the Dartmouth CCMR facility unless otherwise stated. Both male and female mice aged between 8-12 weeks were used for neutrophil isolation experiments.

### Preparation of neutrophils and flow cytometry

Bone marrow was isolated from tibia and femur exudates of 8-12 week old C57BL/6J, *Cftr$^{F508del}$*, CCR2-eGFP$^+$, ASC$^{-/-}$ or Casp1$^{-/-}$ mice. Single-cell suspensions of bone marrow were prepared, and neutrophils were isolated following Miltenyi MACS Neutrophil Isolation Kit. Neutrophils were plated at a density of $2.5 \times 10^5$ in 100µL of OptiMEM in a 96 well plate. The

neutrophils were primed with 100ng/mL O26:B6 *Escherichia coli* LPS (Sigma) and incubated overnight (16h) at 37˚C with 5% $CO_2$. Purity was assessed by flow-cytometry, staining for live C57BL/6J cells using e780, and CD11b and Ly6G for neutrophils. Cell fluorescence was detected by Beckman Coulter Cytoflex S and analyzed using FlowJo software.

## Neutrophil infection assays

Overnight (16-hour) cultures of *P. aeruginosa* were sub-cultured 1:100 in fresh LB high-salt broth supplemented with $MgCl_2$ (10mM) and $CaCl_2$ (0.5mM) and EGTA (5mM) and grown at 37˚C to mid-log phase at $OD_{600}$= 0.5. Cultures were then pelleted, the LB removed, and bacteria were resuspended in PBS to the original volume. Bacterial suspensions were then diluted to an MOI of 10 or 1 in OptiMEM. Overnight media from the neutrophils was removed and replaced with 100μL of OptiMEM containing bacteria. Plates were then incubated at 37˚C with 5% $CO_2$ for 60, 120, or 180 minutes. Cell supernatants (MOI 10) were collected for cytokine ELISAs and lactate dehydrogenase (LDH) assays. Whole wells (MOI 10) were lysed with 1x sample buffer (Invitrogen) containing DTT (Invitrogen), complete mini (Roche) protease inhibitor and PhosSTOP (Roche) phosphatase inhibitor. Cell supernatants were removed (MOI 1), and cells were lysed with 100μL 0.1% NP-40 and serially diluted in PBS and plated on LB agar for CFU assays.

## ExoS secretion assay

Overnight (16-hour) cultures of *P. aeruginosa* were sub-cultured 1:100 in fresh LB high-salt broth supplemented with $MgCl_2$ (10mM) and $CaCl_2$ (0.5mM) and EGTA (5mM) and grown at 37˚C to mid-log phase at $OD_{600}$= 0.5. 1ml of culture was pelleted, supernatant was mixed with trichloro acetic acid (final concentration 10%) and incubated overnight at 4˚C with shaking. Proteins were pelleted by centrifugation, washed with acetone and resuspended in 1x sample buffer containing DTT. Secreted proteins were resolved on SDS page gel and immunoblotted for ExoS.

## Immunoblotting

Cell lysates were run on 4-12% NuPAGE Bis-Tris SDS-PAGE gels (Invitrogen by ThermoFisher Scientific) and transferred to PVDF membranes (ThermoFisher Scientific) using an iBlot 2 Gel Transfer Device (Life Technologies). Membranes were blocked in 5% non-fat dairy milk and incubated with primary antibody overnight. The primary antibodies used were rabbit MAb for GSDMD (Abcam, ab209845), MAb for Histone 3 (Cell Signaling #96C10), Mab for Citrullinated Histone 3 (Abcam #ab5103), MAb for ASC (Cell Signaling #67824), and rabbit polyclonal for β-actin (Cell Signaling, #4967), Mab for Caspase-1 (Adipogen #AG-20B-0042-C100) and rabbit polyclonal antibody for ExoS (from Arne Rietsch). HRP-conjugated anti-rabbit (Jackson Immuno Research) or HRP-conjugated anti-mouse (Jackson Immuno Research) was used as a secondary antibody. Protein bands reacting with antibodies were visualized using chemiluminescent detection reagent (GE Healthcare) on an iBright FL1500 (ThermoFisher Scientific).

## IL-1β quantification

Murine IL-1β in neutrophils supernatants was quantified using ELISA kits (R&D Systems, MLB00C) following the manufacturer's instructions.

## LDH quantification

LDH in neutrophil supernatants was quantified using the CytoTox 96 Non-Radioactive Cytotoxicity Assay (Promega) following the manufacturer's instructions.

## Visualization of nuclear DNA decondensation and incomplete NET Extrusion

BMNs were seeded into glass bottom 96-well plates at a density of $2.5 \times 10^5$ cells/well. Overnight (16h) primed BMNs were infected with *P. aeruginosa* at an MOI of 10 for 60, 120, or 180 minutes. DNA was detected by cell impermeable DNA fluorescent dye (1.0μM Sytox Green) along with cell-permeable DNA dye (1μg/ml Hoechst). Dyes were added 30 minutes before imaging. Fluorescent images were acquired on a 40x objective with a Nikon spinning disc confocal microscope using the Nikon NIS-Elements AR software (version 5.42.02). For automated analysis of DNA staining, we generated a FIJI macro (see S1 Appendix. Fiji macro) to identify cells in each image, count them, and quantify the mean fluorescence intensity of each cell in each channel. Fluorescence values were normalized to an 8-bit scale, and an appropriate threshold set for each channel (5 for Hoechst, 30 for Sytox). Cells positive for Sytox were counted as dead, while those positive for Hoechst alone were counted as live. Data were calculated as the percent of dead cells out of the total cell population. NET formation was assessed by eye by three independent evaluators in the laboratory, and their results were averaged to give the final quantification. Data were calculated as the percent of NET-positive cells out of the total Sytox-positive cell population. Images analyzed per experiment were averaged. Each experiment was performed three times.

## ExsA-ExsD complex modeling

A model of the tertiary structure of the ExSA-ExSD complex was generated using the Alphafold2-multimer tool of ColabFold [44]. The protein sequences of ExSA and ExsD were entered into the Alphafold2-multimer server and run using default settings. PyMOL was used to generate illustrative figures and label the hyperactivating codon changes.

## Bacterial two hybrid

The BACTH system based on reconstitution of adenylate cyclase [45] from Euromedex was used to measure interaction between ExsA and ExsD in *E. coli* BTH101. The *exsA* and *exsA*$^{T48I}$ genes were cloned into pKNT25 and *exsD* was cloned into pUT18. pKNT25-ExsA$^{T48I}$ and pUT18-ExsD were co-transformed into BTH101 to determine interaction. pKNT25-ExsA and pUT18-ExsD or pUT18C-Zip and pKT25-Zip were co-transformed into BTH101 as a positive controls. Negative controls included pUT19-ExsD with empty vector, pKNT25-ExsA with empty vector, pKNT25-ExsAT48I with empty vector, as well as pUT18 with pKNT25 empty vectors. Co-transformants were grown on LB agar containing ampicillin (100 μg/ml) and kanamycin (50 μg/ml) selection at 37 °C. To visualize protein-protein interaction, single colonies were patched onto LB agar containing ampicillin (100 μg/ml) and kanamycin (50 μg/ml) selection with IPTG (0.5mM) and X-Gal (40g/mL) at 30 °C until interaction between positive controls was observed through production of a blue pigment. To quantify the level of the interaction between proteins, co-transformants were plated on LB agar containing ampicillin (100 μg/ml) and kanamycin (50 μg/ml) selection with IPTG (40g/mL) and incubated at 30˚C overnight. Cells were harvested and a β-galactosidase assay was performed following the protocol developed by Battesti et al. [45]. Briefly, harvested cells were normalized to an $OD_{600} =$ 0.5. 100μL of culture are mixed with 900μL of Z-Buffer ($Na_2HPO_4$ (60mM), $NaH_2PO_4.H_2O$ (40mM), KCl (10mM), $MgSO_4.7H_2O$ (1mM), pH 7.0) and permeabilized with chloroform and SDS (0.1%). Cultures were then mixed with 200μL OPNG (4mg/mL in Z-buffer). The reaction was stopped with the addition of $Na_2CO_3$ (1M). Samples were measured at OD at 420nm and 550nm and values calculated as follows: Miller Units = $1000*[(OD_{420}-1.75*OD_{550})/ (time in minutes*OD_{600}*volume of culture used)]$. In this case $OD_{600}=0.5$ and volume was 100μL, time in minutes was measured as reaction time after addition of OPNG to addition of $Na_2CO$.

## Statistical analysis of data

Experimental data generated from at least three independent experiments are analyzed for significance using GraphPad Prism software. Probability ($p$) values for ELISA and LDH data are calculated using one-way analysis of variance (ANOVA) or grouped two-way ANOVA with Tukey's posttest. P values of less than or equal to 0.05 were considered significant.

## Supporting information

**S1 Table. Primers.**
(PDF)

**S1 Appendix. Fiji macro.**
(PDF)

**S1 Fig. Characterization of BMNs primed with LPS by flow cytometry.** BMNs were isolated from B6 mice and analyzed immediately without staining (A), immediately with staining (B) or after 18 hr incubation without (C) or with (D) 100 ng/ml LPS. BMNs were stained with e780, Ly6G-PE and CD11b-FITC and analyzed by flow cytometry. (A-D) The gates indicate from left to right: singlets, normal size, live cells, and cells that are Ly6G$^+$ and CD11b$^+$. Histograms show overlay of fluorescence intensity for Ly6G$^+$ (E) or CD11b$^+$ (F) in each group.
(PDF)

**S2 Fig. Analysis of BMN infections with PAO1F or ExoS and/or ExoT ADPRT catalytic mutants.** B6 BMNs were left UI or infected for 60 min with PAO1F, ExoS(A-), ExoT(A-), or the ExoS(A-)ExoT(A-) double mutant at MOI 10 and analyzed for released IL-1β (A) or LDH (B). Data represent normalized values for $2.5 \times 10^5$ cells/well ± the standard deviation from three independent experiments. Significant differences were determined by two-way ANOVA comparing to PAO1F. ns, not significant; * P<0.05; ** P<0.01.
(PDF)

**S3 Fig. Analysis of ExoS secreted by *P. aeruginosa* isolates.** The indicated strains of *P. aeruginosa* were grown in bacterial media that activates the T3SS. Secreted ExoS was collected from supernatants and detected by immunoblotting using anti-ExoS antibody. In (A) indicated strains of PAO1F and p32 isolates were analyzed. In (B) left lower ExsD$^{WT}$ and left upper ExsD$^{T188P}$ strains were analyzed. In (C) three independent p32_08 ExsA$^{T48I}$ and two independent p32_85 ExsA$^{WT}$ allele swap mutants were analyzed along with one parental control (WT or T48I) strain for each.
(PDF)

**S4 Fig. Analysis of BMN infections with PAO1F strains or patient 32 isolates.** B6 BMNs were left UI or infected for 60 min with laboratory strains PAO1F or its T3SS null mutant Δ*pscD* or p32 isolates p32_08, p32_85, p32_86, or p32_108 at MOI 10 (A, B) and analyzed for released IL-1β (A) or LDH (B). Data represent normalized values for $2.5 \times 10^5$ cells/well ± the standard deviation from 3 independent experiments. Significant differences were determined by one-way ANOVA comparing to PAO1F for UI or Δ*pscD,* or comparing to p32_08 for other patient 32 isolates, or comparing between conditions as shown by brackets. ns, not significant; * P<0.05; ** P<0.01; *** P<0.001.
(PDF)

**S5 Fig. Analysis of BMN infections with patient 32 isolates with ExsA allele swaps.** B6 BMNs were left UI or infected for 60 min with p32 isolates p32_08, p32_85, or their allele swapped ExsA variants and analyzed for released IL-1β (A) or LDH (B). Data represent

normalized values for $2.5 \times 10^5$ cells/well ± the standard deviation from 3 independent experiments. Significant differences were determined by one-way ANOVA comparing to p32_08 ExsA[WT] or comparing between conditions as shown by brackets. ns, not significant; * P<0.05; ** P<0.01; *** P<0.001.
(PDF)

**S6 Fig. AlphaFold2 prediction of ExsA-ExsD complex and bacterial two hybrid assay of ExsA-ExsD interaction.** (A) ExsA is shown in green with position of T48I. ExsD is shown in magenta with positions of T188P [24] and S164P [23]. (B) *E. coli* B2H reporter strains containing pairs of vectors encoding the indicated Exs or Zip proteins or empty vectors (EV) were assayed for production of β-galatosidase as measured in Miller Units. Each data point is from an independent experiment. Significant differences as compared to ExsA+ExsD was determined by one-way ANOVA. ns, not significant; **** P<0.0001.
(PDF)

**S7 Fig. Model of neutrophil responses to hyperactive T3SS ExoS+ or ExoS ADPRT- P. aeruginosa.** (A) Hyperactive T3SS ExoS+ *P. aeruginosa* translocates flagellin and active ExoS into neutrophil. ExoS inhibits phagocytic signaling by targeting host proteins such as Ras. ExoS inhibits assembly of NLRC4 inflammasome in response to flagellin detection by NAIP. ExoS promotes formation of unknown pores in plasma membrane, allowing influx of $Ca^{2+}$. $Ca^{2+}$ enters the nucleus to activate Pad4, resulting in Cit3 generation, chromatin decondensation into cytosol, and incomplete NET extrusion due to maintenance of the cortical actin cytoskeleton. Pore formation triggers NINJ1 activation, PMR and release of LDH and HMGB1. Glycine inhibits activation of NINJ1 to prevent PMR. (B) Hyperactive T3SS ExoS ADPRT- *P. aeruginosa* translocates flagellin and inactive ExoS into neutrophil. ExoS ADPRT- fails to inhibit phagocytic signaling and assembly of NLRC4 inflammasome. ExoS ADPRT- fails to promote pore formation and unleashes caspase-1 to cleave pro-IL-1β and GSDMD. N-GSDMD forms plasma membrane pores, allowing $Ca^{2+}$ to enter the nucleus to activate Pad4, resulting in Cit3 generation, chromatin decondensation into cytosol, and incomplete NET extrusion. GSDMD pore formation triggers NINJ1 activation, PMR and release of LDH and HMGB1. Glycine inhibits activation of NINJ1 to prevent PMR.
(PDF)

## Acknowledgments

We acknowledge the important contributions of Natasha Medici to initiating study of ExoS and ExoT in the laboratory. We thank Youssef Aachoui for feedback on the manuscript, Arne Rietsch for the ExoS antibody, and Arne Rietsch, Jennifer Bomberger and Pradeep Singh for *P. aeruginosa* strains, and Joshua Obar for Asc-/- mice. We thank Ko-Wei Liu and Alexander Rapp for assistance with acquisition and analysis of flow cytometry data, and Joseph Pennington for help with NET analysis. We thank the Dartmouth Life Sciences Light Microscopy facility which is supported by bioMT through NIH grant P20-GM113132. We acknowledge Dartmouth Genomics and Molecular Biology Shared Resources Core Facility (NCI Cancer Center, Support Grant 5P30CA023108-37). Flow cytometry experiments were carried out in DartLab, the Immune Monitoring and Flow Cytometry Shared Resource at the Norris Cotton Cancer Center at Dartmouth, with NCI Cancer Center Support Grant 5P30 CA023108-41 and COBRE grant P30GM103415-15.

## Author contributions

**Conceptualization:** Arianna D. Reuven, James B. Bliska.

**Data curation:** Arianna D. Reuven, Sarah Katzenell.

**Formal analysis:** Arianna D. Reuven, Sarah Katzenell, Bethany W. Mwaura.

**Funding acquisition:** James B. Bliska.

**Investigation:** Arianna D. Reuven.

**Methodology:** Arianna D. Reuven, Sarah Katzenell, Bethany W. Mwaura, James B. Bliska.

**Project administration:** James B. Bliska.

**Resources:** James B. Bliska.

**Software:** Sarah Katzenell.

**Supervision:** James B. Bliska.

**Validation:** Arianna D. Reuven.

**Visualization:** Bethany W. Mwaura.

**Writing – original draft:** Arianna D. Reuven, James B. Bliska.

**Writing – review & editing:** Arianna D. Reuven, James B. Bliska.

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
