## [Decision Letter · Decision Letter 0]

20 Nov 2024

PPATHOGENS-D-24-02182ExoS Effector in Pseudomonas aeruginosa Hyperactive Type III Secretion System Mutant Promotes Enhanced Plasma Membrane Rupture in NeutrophilsPLOS Pathogens

Dear Dr. Bliska,

Thank you for submitting your manuscript to PLOS Pathogens. After careful consideration, we feel that it has merit but does not fully meet PLOS Pathogens's publication criteria as it currently stands. Therefore, we invite you to submit a revised version of the manuscript that addresses the points raised during the review process.

Please submit your revised manuscript within 60 days Jan 19 2025 11:59PM. If you will need more time than this to complete your revisions, please reply to this message or contact the journal office at plospathogens@plos.org. Please include the following items when submitting your revised manuscript:

* A rebuttal letter that responds to each point raised by the editor and reviewer(s). You should upload this letter as a separate file labeled 'Response to Reviewers '. This file does not need to include responses to any formatting updates and technical items listed in the 'Journal Requirements' section below. * A marked-up copy of your manuscript that highlights changes made to the original version. You should upload this as a separate file labeled 'Revised Manuscript with Track Changes '. * An unmarked version of your revised paper without tracked changes. You should upload this as a separate file labeled 'Manuscript '. If you would like to make changes to your financial disclosure, competing interests statement, or data availability statement, please make these updates within the submission form at the time of resubmission. Guidelines for resubmitting your figure files are available below the reviewer comments at the end of this letter. We look forward to receiving your revised manuscript.  Kind regards,Vincent T LeeAcademic EditorPLOS Pathogens

Matthew Wolfgang

Section Editor

PLOS Pathogens

Michael Malim

Editor-in-Chief

PLOS Pathogens

orcid.org/0000-0002-7699-2064

**Additional Editor Comments:**

The reviewers have found that the evidence largely support the title of the manuscript. However, reviewers #2 and 3 found that there are possible issues with the clinical strains that go beyond T3SS. In addition to addressing their specific comments, the addition of a mutant lacking T3SS would greatly support the effect is solely due to T3SS and ExoS.

**Journal Requirements:**

2) Please provide an Author Summary. This should appear in your manuscript between the Abstract (if applicable) and the Introduction, and should be 150u2013200 words long. The aim should be to make your findings accessible to a wide audience that includes both scientists and non-scientists. Sample summaries can be found on our website under Submission Guidelines:

https://journals.plos.org/plospathogens/s/submission-guidelines#loc-parts-of-a-submission

- ® on Lines: 519, 521, and 522.

5) We have noticed that you have uploaded Supporting Information files, but you have not included a list of legends. Please add a full list of legends for your Supporting Information files after the references list.

6) We note that your Data Availability Statement is currently as follows: "The data used in this submission can be accessed in the supporting information files.". Please confirm at this time whether or not your submission contains all raw data required to replicate the results of your study. Authors must share the “minimal data set” for their submission. PLOS defines the minimal data set to consist of the data required to replicate all study findings reported in the article, as well as related metadata and methods (https://journals.plos.org/plosone/s/data-availability#loc-minimal-data-set-definition).

- The points extracted from images for analysis..

7) Please amend your detailed Financial Disclosure statement. This is published with the article. It must therefore be completed in full sentences and contain the exact wording you wish to be published.

**Reviewers' Comments:**

Reviewer's Responses to Questions

**Part I - Summary**

Reviewer #1: The manuscript by Reuven et al. carefully examines the interaction of P. aeruginosa strains producing they type III secreted effector ExoS with neutrophils. Importantly for researchers in the cystic fibrosis arena, they both examine the interaction with bone-marrow derived neutrophils from wild type and deltaF508 mice. They also examine the effect of an exsD mutant that results in up-regulation of the T3SS and was identified in a CF patient isolate, further increasing the relevance for this patient group. The paper slots in nicely with a few recent reports, all cited in the manuscript, that analyze inflammasome activation in neutrophils infected by PAO1. The authors do a more careful analysis of the chromatin decondensation/induction of NETosis, and also expand on the role of ninjurin1 in plasma membrane rupture that occurs upon infection. In the end, the authors present evidence for three processes: 1) GSDM-dependent, ExoS-inhibited, NINJ1-independent IL1-beta release, 2) ExoS-ADPR-independent, but secretion-activity regulated, NINJ1-dependent PMR, and 3) ExoS-ADPR and NINJ1 independent, but secretion-activity regulated H3 citrulination. This, in conjunction with work by other labs that infer a role for NLRP3 in some of these processes begins to show the complicated nature of the interaction of P. aeruginosa with this key immune cell and helps focus future research trying to unravel these related processes and hopefully put them in the context of PA pathogenesis. The experiments were generally well-presented and explained.

Reviewer #2: The manuscript explores the role of ExoS of hyperactive T3SS Pseudomonas aeruginosa mutants in promoting neutrophil plasma membrane rupture and incomplete neutrophil extracellular trap (NET) extrusion via NLRC4-independent manner. The study utilizes bone marrow derived murine neutrophils along with clinical isolates obtained from people with cystic fibrosis (CF) and laboratory strain PAO1F. Using LPS-primed neutrophils, the authors have demonstrated that ADPRT activity of ExoS suppresses IL-1β release by neutrophils challenged with PAO1F. The authors have also identified a CF isolate that is a hyperactive T3SS strain due to a mutation in exsA, a transcription factor of ExoS. The hyperactive T3SS mutant promotes citrullination of histone H3, nuclear decondensation, incomplete NET extrusion and plasma membrane rupture. The authors have further demonstrated that plasma membrane rupture occurs independent of caspase-1 activation as well as inflammasome adaptor ASC and instead is a result ninjurin. Overall, the manuscript is well written and addressing an important topic. There are some issues raised below that should be considered prior to publication.

Reviewer #3: The interaction of Pseudomonas aeruginosa with neutrophils is an important topic of investigation allowing us to further understand the pathogenesis of infection and pathology in the airways and other infection sites. The authors used a hyperactive type III secretion mutant isolated from a CF patient to investigate the effects of ExoS ADPR activity on inflammasome activation and plasma membrane rupture (PMR) in murine neutrophils with and without inflammasome components (mutants in caspase 1 or ASC) and compared wild-type with engineered CF neutrophils with delta F508 mutation. Results appeared to be confirmatory of multiple prior studies of ExoS effects on neutrophil function and inflammasome activity but showed that ExoS effects on PMR (cytotoxicity LDH release) were relatively independent from inflammasome (IL-1beta) changes throughout the study. PMR was inhibited by glycine suggesting involvement of injurin-1 in PMR.

The strengths of the study are the genetic analysis of the mechanism for T3SS hyper-secretion in the CF mutant, the consistent delineation of bacterial ExoS mediated PMR from effects on inflammasome modulation throughout the study, and the likely mechanism for PMR involving ninjurin-1. The manuscript was well written and thoughtfully discussed. Strong execution and scholarship consistent with this excellent research group.

Weaknesses per se could be the reliance on two clonal isolates from one patient, the lack of a defined exsD mutant control, unclear as to bacterial location (intracellular vs. extracellular) when interacting with neutrophils and absence of imaging quantification in Figs. 5 and 6. Also, it appeared that some of the activity of ExoS attributed to ADPr activity was similar with ExoS ADFPr mutant, e.g. Fig. 8B.

**Part II – Major Issues: Key Experiments Required for Acceptance**

Reviewer #1: n/a

Reviewer #2: L141-143, L307-309: The authors claim that the ADPRT activity of ExoS is necessary for PAO1 to inhibit NLRC4 inflammasome activation and IL-1β release. While the data in Fig. 2A demonstrates the latter, there is no direct evidence in Fig. 2 that demonstrates inhibition of NLRC4 inflammasome activation. It would seem important to include data demonstrating that NLRC4 is no longer inhibited by a catalytically inactive ExoS mutant to support the authors’ claim. There should also be some discussion regarding the PMR data in Fig. S2B with respect to the ExoT mutant effect. Was this expected?

L148. The results in Fig. 2 indicate that the CFTR genotype does not appear to be involved in IL-1β release from neutrophils. The rationale in the next section was to increase the biological significance of the results so additional CF isolates were investigated. These seem at odds with each other.

L176-183. Granted this likely took some time to accomplish these experiments but I’m not certain this adds anything additional to the main message of the paper. What experiments proposed in lines 181-183 would be conducted to get at this question?

L185/196. Have the genome sequences of these hyperactive mutants been determined? Are the differences between these strains only due to the T48I mutation in exsA? This is partially addressed in Fig. S3B with cross complementation, but these studies evaluated ExoS levels. Might there be alterations outside exsA that could account for the data in Fig. 3/4?

L207-209: The authors state that citH3 levels were elevated in neutrophils infected with p32_85 (hyperactive T3SS) compared to p32_08. The result is puzzling as neutrophils infected with the catalytic mutant ExoS(A-) also exhibits increased citH3. It is likely that the clinical isolate p32_85 may harbor additional pathoadaptive mutations in addition to that in exsA. Related to the above point, can the authors comment on whether p32_85 is genetically identical to PAO1F with the exception of exsA? Alternatively, constructing a hyperactive T3SS mutant in PAO1F background and challenging the neutrophils to examine whether increased citH3 levels are indeed a result of the ExoS ADPRT activity in the hyperactive T3SS mutant and is independent of NLRC4 activation.

Fig.4. Quantification of immunoblot depicting levels of H3 and citH3 is required as the H3 signal for ExoS(A-) lane is more intense than the other H3 signals. This makes it difficult to interpret the higher signal intensity of citH3 for ExoS(A-) lane. Quantification of citH3 signal normalized to H3 signal is recommended. This analysis can be extended to other components of the blot to strengthen the observations.

Fig.6. Quantification of the microscopy images depicting chromatin decondensation and NET extrusion is required to support the statement in L213-214 that hyperactive T3SS mutant P. aeruginosa promotes nuclear decondensation and incomplete NET extrusion in BMNs. It is difficult to distinguish if there are differences in these events between a ExoS (A-) strain and a hyperactive T3SS strain.

L253-254: The authors conclude from Figure 7 that the ExoS ADPRT is promoting PMR using LDH assay. In Fig. 8B, when using the hyperactive T3SS strain p32_85 and its corresponding ExoS (A-) mutant, the LDH released by neutrophils challenged by both strains is comparable, suggesting that ExoS ADPRT is not involved in PMR when looking at this hyperactive T3SS strain.

L264-265: Although it is reported that ExoS ADPRT activity is necessary to inhibit NLRC4 activation, the figure corresponding to this data (Fig. 8A, C) shows increased IL-1β release and cleavage of Gasdermin D, both of which are not exclusive to NLRC4 activation. Direct evidence using an anti-NLRC4 for immunoblot would seem to be required to support the statement.

Fig 8 and 9. PAO1F and ExoS (A-) control strains seem to be missing from the experiments. Since the experiments for these figures use clinical isolates that are resulting in unexpected results as stated by the authors in L269-271, it would seem pertinent to use laboratory WT strain and its mutant to validate these observations. Furthermore, in L269-271, the authors compare the observations of laboratory parent strain PAO1F and its ExoS (A-) mutant in previous figures (Fig.4 and 7) of the manuscript which further demonstrates the need for including these control strains in the experiments for Figures 8 and 9 for comparison.

Reviewer #3: 1. Were the clonal isolates from patient 32 examined for defects or hyperactivity in any other virulence mechanisms/factors? Was there only T3SS hyper-secretion?

2. The imaging in Figs. 5 and 6 could be quantified with statistical analysis for clearer representation of the findings?

3. Was the bacterial location determined after neutrophil interaction? It was unclear if they were inside or outside of cells? A number of comments suggested they were internalized as might be expected.

4. Was a defined exsD mutant included at any time for comparison with the T3SS hyper-secretion clonal mutant?

5. In Fig. 8B, the PMR (LDH) appeared similar in clonal isolate P32-85 and its ExoS ADPr mutant. This seems to contradict statement that ExoS ADPr activity is driving the process.

**Part III – Minor Issues: Editorial and Data Presentation Modifications**

Reviewer #1: Line 209 – could this be due to ExoS-dependent activation of NLRP3 (which seems to occur concomitantly with NLRC4 inactivation in neutrophils)?

Line 232 – sorry for being the non-immunologist idiot here. While I can see the puncta, I’m not sure I understand the explanation here. How does relaxed chromatin that is confined with in the BMN (small, bright foci) differ from decondensation without NET extrusion (cell-filling DNA stain)?

Fig. 7B. How come LDH release is up in the p32_85-infected cells that hypersecrete ExoS, when ExoS seems to interfere with LDH release in the case of the PAO1F strain?

Fig. 9. How specific is the glycine-mediated inhibition for Ninjurin-mediated plasma membrane rupture? The authors propose doing the experiments with ninjurin null cells as a follow-up experiment, which is reasonable, but I wonder if there is any information on this in the literature that can be cited to add perspective?

297 (and 379?) - Could membrane-perturbation by translocon pore insertion combined with a ExoS-mediated inhibition of membrane damage repair be the trigger for NINJ1-dependent cell lysis? Control of H3 citrulination, however, would have to be controlled in a NINJ1 independent manner, perhaps by influx of calcium via the translocon. This may also relate to greater translocon insertion in the cells infected with the hyperactive mutant.

Reviewer #2: Suggest replacing “CF patients” to individuals/people with CF wherever applicable (L35, 86, 150, 317). CFF is requesting this in publications and presentations.

Fig 5 and 6: Suggest including scale bars in figure legend as the text is too small to read in the microscopy images.

Reviewer #3: The abstract could be clearer in representing the findings unique to the current study. Much of the abstract was dedicated to describing background studies rather than novel results.

PLOS authors have the option to publish the peer review history of their article (what does this mean? ). If published, this will include your full peer review and any attached files.

**Do you want your identity to be public for this peer review?** For information about this choice, including consent withdrawal, please see our Privacy Policy .

Reviewer #1: No

Reviewer #2: No

Reviewer #3: No

**Figure resubmission:**
---

## [Editor Report · Decision Letter 1]

5 Mar 2025

Dear Prof. Bliska,

We are pleased to inform you that your manuscript 'ExoS Effector in Pseudomonas aeruginosa Hyperactive Type III Secretion System Mutant Promotes Enhanced Plasma Membrane Rupture in Neutrophils' has been provisionally accepted for publication in PLOS Pathogens.

Best regards,

Vincent T Lee

Academic Editor

PLOS Pathogens

Matthew Wolfgang

Section Editor

PLOS Pathogens

Sumita Bhaduri-McIntosh

Editor-in-Chief

PLOS Pathogens

orcid.org/0000-0003-2946-9497

Michael Malim

Editor-in-Chief

PLOS Pathogens

orcid.org/0000-0002-7699-2064

Thank you for taking the reviewers suggestions seriously and modifying the manuscript extensively to address their concerns. Congratulations.